

**FORUM paper: The significance of soils and soil science towards realization of the UN sustainable**
**development goals (SDGs).**
Keesstra, S.D.[1], Bouma, J.[2], Wallinga, J.[3], Tittonell, P.[4], Smith, P.[5], Cerdà A.[6], Montanarella, L.[7],
Quinton, J.[8], Pachepsky, Y.[9], van der Putten,[10,11] W.H, Bardgett, R.D[12], Moolenaar, S.[13], Mol, G.[14],
Fresco, L.O.[15]
[1] Soil Physics and Land Management Group, Wageningen University, The Netherlands, saskia.keesstra@wur.nl
[2] Formerly Soils Department, Wageningen University. Johan.bouma@planet.nl
[3] Soil Geography and Landscape Group, Wageningen University, The Netherlands
[4] Natural Resources and Environment Program, Instituto Nacional de Tecnología Agropecuaria (INTA),
Argentina
[5] University of Aberdeen, Institute of Biological and Environmental Sciences, Aberdeen, United Kingdom
[6] Departament de Geografia. Universitat de València. Blasco Ibàñez, 28, 46010-Valencia. Spain
[7] European Commission, Joint Research Centre, Italy
[8] Lancaster University, Lancaster Environment Centre, Lancaster, United Kingdom, j.quinton@lancaster.ac.uk
[9] USDA-ARS, Environmental Microbial and Food Safety Laboratory, Beltsville Agricultural Research Center,
Beltsville, MD, United States
[10] Department of Terrestrial Ecology, Netherlands Institute of Ecology NIOO-KNAW, Droevendaalsesteeg 10,
Wageningen, NL- 6708, The Netherlands.
[11] Department of Nematology, Wageningen University, Droevendaalsesteeg 1, 6708 PB, Wageningen, NL- 6708,
The Netherlands
[12] University of Manchester, Faculty of Life Sciences, Manchester, United Kingdom
[13] Commonland, Department of Science & Education, www.commonland.com, Amsterdam, The Netherlands
[14] Alterra, Wageningen University and Research Centre, Wageningen, The Netherlands
[15] Wageningen University and Research Centre, Wageningen, The Netherlands
**Abstract**
In this FORUM paper we discuss how soil scientists can help to reach the recently adopted UN
Sustainable Development Goals in the most effective manner. Soil science, as a land-related
discipline has important links to several of the SDGs which are demonstrated through the functions
of soils and the ecosystem services that are linked to those functions. We explore and discuss how





soil scientists can rise to the challenge both internally, in terms of our procedures and practices, and
externally in terms of our relations with colleague scientists in other disciplines, diverse groups of
stakeholders and the policy arena. To meet these goals we recommend the following steps to be
taken by the soil science community as a whole: (i) Embrace the UN Sustainable Development Goals,
as they provide a platform that allows soil science to demonstrate its relevance for realizing a
sustainable society by 2030. (ii) Show the specific value of soil science: Research should explicitly
show how using modern soil information can improve the results of inter- and trans-disciplinary
studies on SDGs related to food security, water scarcity, climate change, biodiversity loss and health
threats. (iii) Given the integrative nature of soils, soil scientists are in a unique position to take
leadership in overarching systems-analyses of ecosystems; (iii) Raise awareness of soil organic matter
as a key attribute of soils to illustrate its importance for soil functions and ecosystem services; (iv)
Improve the transfer of knowledge through knowledge brokers with a soil background; (v) Start at
the basis: educational programs are needed at all levels, starting in primary schools, and emphasizing
practical, down-to-earth examples; (vi) Facilitate communication with the policy arena by framing
research in terms that resonate with politicians in terms of the policy cycle or by considering drivers,
pressures and responses affecting impacts of land use change; and finally (vii) all this is only possible
if researchers, with soil scientists in the frontlines, look over the hedge towards other disciplines, to
the world-at-large and to the policy arena, reaching over to listen first, as a basis for genuine
collaboration.

## 1. Introduction: what is the challenge?

In this FORUM paper we discuss how the soil science profession can address the challenges of the
recently adopted UN Sustainable Development Goals in the most effective manner. The sustainability
of human societies depends on the wise use of natural resources. Soils contribute to basic human
needs like food, clean water, and clean air, and are a major carrier for biodiversity. In the globalized
world of the 21$^{st}$ century, soil sustainability not only depends on management choices by farmers,
foresters and land planners but also on political decisions on rules and regulations, marketing and
subsidies, while public perceptions are perhaps the most important issue. The United Nations have
proposed seventeen sustainable development goals, which not only present a clear challenge to
national governments but also to a wide range of stakeholders. Montanarella and Lobos Alva (2015)
have provided a historical description of the way in which soils have been discussed in UN documents
in recent decades. Their paper demonstrates that, even though soils are essential to sustainable
development, they have never been the specific focus of a Multilateral Environmental Agreement
(MEA). However, as a crosscutting theme soils are considered within the three "Rio Conventions"

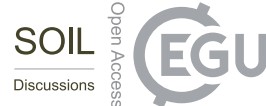

negotiated at the United Nations Conference on Environment and Development (UNCED) in Rio de
Janeiro in 1992: (i) the United Nations Framework Convention on Climate Change (UNFCCC); (ii) the
United Nations Convention on Biological Diversity (CBD) and (iii) the United Nations Convention to
Combat Desertification (UNCCD). As the main binding global environmental agreements these "Rio
Conventions" are considered the framework in which individual countries can implement sustainable
development initiatives, aiming at the mitigation of human induced climate change, the protection of
biological diversity and the limitation of desertification processes in drylands.
Soils play an important role in each of these issues. Putting soils on the agenda of these MEAs has
involved a long process that required a large effort of awareness-raising and communication of issues
related to the degradation of soils and land by scientists, civil society organizations and policy-
makers. In spite of these efforts, the convention texts of CBD and UNFCCC do not explicitly discuss
the crucial role of soils. In contrast, soils are addressed in the convention text of the UNCCD, but only
restricted to drylands, and in actions prescribed by the three conventions. These actions include the
development of national action plans and the definition of specific targets and indicators for the
monitoring of natural resources at national level. Twenty years after the conference in Rio, the
achievements were analysed at the Rio+20 meeting on sustainable development in 2012 in Rio de
Janeiro. This analysis showed that some progress has been made, but that extensive land and soil
degradation still occur all over the world and fertile soil resources are still rapidly depleted, reducing
the potential for food production. Conscious of these alarming trends, countries participating at the
Rio+20 sustainable development conference agreed in the outcome document "The Future We
Want" that we should "*strive to achieve a land degradation-neutral world in the context of*
*sustainable development*" (Mueller and Weigelt, 2015). This agreement was further developed
during the subsequent process to define Sustainable Development Goals (SDGs), approved by the UN
General Assembly in September 2015 (Table 1). This soft-law process reflects the growing interest in
the development of a universal and transformative agenda that provides a global vision for
sustainable development, linking environmental, economic and societal issues. Main difference to
the previous Millennium Development Goals (MDG) is that the SDGs are applicable by all countries in
the world, not just by developing countries. Every Nation has to implement now these goals in order
to achieve by 2030 the agreed targets.
Every scientific discipline faces the challenge to act upon these SDGs and this is particularly relevant
for soil science, as a land-related discipline with important links to several of the SDGs. In this FORUM
Paper we explore and discuss how soil scientists can rise to the challenge both internally in terms of
our procedures and practices and externally in terms of our relations with colleague scientists in
other disciplines, diverse groups of stakeholders and the policy arena.




### 2. **Addressing the Sustainable Development Goals.**

The broad Sustainable Development Goals (Table 1) are intended to be a guideline for all governments. Some Goals are mainly socio-economic in character (e.g. Goals 1,4,5,8,9,10,11,16,17) while others focus clearly on the biophysical system, in which soils play a clear role (e.g. Goals 2,3,6,7,12,13,14,15). Although it is tempting to make the distinction between a focus on socio-economics and on the biophysical system, these two realms together define human existence and mutually depend on each other. For achieving goals with a socio-economic focus we need to consider the associated dynamic behaviour of ecosystems while for achieving goals with an ecosystem focus, we need to consider socio-economic aspects. Environmental sustainability will depend on the actions of land users such as, for example, farmers and forest managers, but also urban developments have major effects on local land use. The SDGs not only present a real challenge to the citizens of the world and their various policy arenas. The scientific community has a responsibility to provide all stakeholders with information that allows them to make informed choices. We believe that the introduction of SDGs in the international YEAR OF SOILS 2015 offers a new and unique opportunity for the soil science community to show that soil science can make significant contributions to several of the SDGs. Although this notion is clearly growing, we feel that a well-focused action is needed to urge fellow (soil) scientists, members of the policy arena and stakeholders and citizens at large, to act according to this notion. Actions needed are different for each of these groups; in this FORUM paper we will focus on the implications for actions by the soil science community. Important educational efforts for stakeholders and the public at large, with particular attention for primary education of children, have been addressed elsewhere (Bouma et al., 2012).

It is important to recognize that for most SDGs, there is no direct link with soils. Rather, soils contribute to general ecosystem services, defined as "services to society that ecosystems provide" which requires cooperation between different disciplines (e.g. De Groot et al., 2002; Dominati et al., 2014; Robinson et al., 2014). Ecosystem Services contribute to nearly all land-related SDGs, either directly or indirectly. Table 1 shows ecosystem services as they are now recognized in the soil literature (e.g. Dominati et al., 2014). The question can be raised as to how input of soil expertise can be most effective when defining ecosystem services. A logical way to consider soil contributions to interdisciplinary studies on ecosystem services is to consider the seven soil functions, as defined by the European Commission (EC, 2006) (Table 2). Thus, an operational sequence is defined starting with the SDGs, next considering relevant ecosystem services and the contributions that the soils can make to improve those services (see also Fig. 1). Most applied soil studies can be expressed in terms of their relevance for certain SDGs, also indicating which ecosystem services and associated soil





functions play an important role. This new possibility for framing soil studies, offers an opportunity to
increase the visibility and recognition of the work in soil science as a much wider audience is being
addressed. Bouma et al. (2015) illustrated this reframing process for six published studies on soil and
water management in the Netherlands and Italy.
A clear framework linking SDGs, ecosystem services and soil functions will also pave the way towards
a more relevant contribution of the soil science community to on-going major global and regional
ecosystems assessments related to land and soils. The most obvious example is the currently on-
going Land Degradation and Restoration Assessment (LDRA) of the Intergovernmental Platform for
Biodiversity and Ecosystem Services (IPBES), planned for final release in early 2018. Similar to IPCC,
these assessments by IPBES will be the main scientific reference for future policy development on
terrestrial ecosystems at global, regional and national scale.
Overall, we should acknowledge that services are provided by nature, and that human efforts should
be governed by the realisation that every ecosystem has its own, characteristic dynamics and
thresholds. Sustainable development can only be achieved when taking into account processes,
feedbacks and thresholds in the eco-system.
In summary, the aim of this FORUM Paper, is therefore to discuss how soils can contribute to the
realization of the SDGs. We urge soil scientists to pursue a central role in the system analysis
approach that is needed to confront the societal challenges of our time. For this we argue why soil
scientists need to reach out to other scientific disciplines, and to stakeholders outside of science.
Awareness raising on all levels in society will play a key role in this. Six short essays, written by invited
experts expressing their personal impressions, feature prominently in this FORUM paper, and serve
to introduce the discussion, covering key issues for soil science that are also part of several of the
SDGs: food, health, water, climate and land management. This paper also serves as an introductory
FORUM paper to this Special Issue on "Soil Science in a Changing World", which contains selected
contributions of participants of the Wageningen Soil Conference (Wageningen, August 2015), and
EGU Union Symposium: Soil Science within an interdisciplinary framework (Vienna, April 2015).

**3. The six main issues:**

**Essay 1: Food security (SDGs 1, 2 and 3): Soil fertility and the role of soils for food security in developing countries**

Addressing current and future food security is not just a matter of producing more food globally. Agricultural productivity must increase where food is most needed, and where both rural and urban populations are expected to increase the fastest in the near future. This is the situation in most of sub-Saharan Africa and in several other regions of Latin America, Asia and the Pacific (UNDESA, 2013). There are some common denominators to these regions. In the first place, the inability of the majority of smallholder farmers to access and/or to afford agricultural inputs (Pretty et al., 2011, Tittonell, 2014). Second, the severity with which climate



change impacts on some of these regions (Thornton et al., 2014). Third, the extent of soil degradation, which is estimated at 25% of the arable land in the world (Vlek et al., 2008). And finally, the fact that some of these regions are hosting valuable biodiversity and/or delivering ecosystem services of global or regional importance, (Hooper et al., 2005)which often leads to competing claims between local and international communities.

It has been repeatedly shown that the technologies of industrial agriculture as practiced in developed regions are ineffective at sustaining soil productivity in the context of smallholder family agriculture (Tittonell and Giller, 2013). Restoring soil productivity and ecosystem functions in these contexts requires new ways of managing soil fertility.

These include:

(i) innovative forms of 'precision' agriculture that consider the diversity, heterogeneity and dynamics of smallholder farming systems; precision agriculture implies more than just using GPS;, it is also about targeting resources in space and time to increase efficiency, build resilience and reduce negative impacts; local knowledge can be used as the basis for precision agriculture in developing countries.

(ii) a systems approach to nutrient acquisition and management; agronomy has traditionally addressed the problem of crop nutrition by thinking and acting at the scale of individual fields, and often looking at single resource groups; yet nutrient management cannot be decoupled from management of other farm resources and processes such as recycling are crucial to overall systems efficiency.

(iii) agro ecological strategies for the restoration of degraded soils and the maintenance of soil physical properties; rural population growth in tropical regions of developing countries is leading to accelerated soil degradation, as more land previously under forest or grazing use is brought into annual cultivation; less land available per household prevents soil maintenance practices such as fallow or pasture rotations, leading to greater frequency of soil ploughing and less organic matter inputs; strategies are needed to restore degraded soils and halt current degradation processes in precious land to produce food; but this also requires new institutional arrangement around land tenure and collective resource management. This may involve a large-scale approach, involving multi-stakeholder partnerships built on new business models and sustainable business cases with multiple returns from sustainable land management and landscape restoration. (Ferwerda, 2015).

(iv) to capitalize on the recent and growing understanding on the soil food web to increase nutrient and water use efficiency; the association between nutrient capture and retention in soils and trophic network topologies points to promising avenues towards the design of more efficient and resilience cropping systems; management systems that rely on greater diversity such as agroforestry and intercropping lead to greater diversity of soil organisms and a range of hypothesis on how this can contribute to improve agricultural sustainability are being put forward.


**Essay 2. Health (SDG 3): Soil and public health: a vital nexus.**

Throughout the history of civilization, relationships between soils and human health have inspired spiritual movements, philosophical systems, cultural exchanges, and interdisciplinary interactions, and provided medicinal substances of paramount impact. Modern public health in its efforts on preventing disease, prolonging life and putting health through organized activities and informed choices of society faces the need of understanding and managing interactions between soils and health. Given the climate, resource, and population pressures, such understanding becomes an imperative. Soils sustain life. They affect human health via quantity, quality, and safety of available food and water, as a source of essential medicines, and via direct exposure of individuals to soils.

We are witnessing a paradigm shift from recognizing and yet disregarding the 'soil-health' nexus complexity to parameterizing this complexity and identifying reliable controls. This becomes possible with the advent of modern research tools as a source of 'big data' on multivariate nonlinear soil systems and the multiplicity of health metrics. These advances, in particular, have enabled the demonstration of the dependence of human pathogen suppression in soils and plants on the soil microbial community structure which, in turn, is directly affected by the soil-plant system management (Vivant et al., 2013; Gu et al., 2013).Soil eutrophication appears to create favourable conditions for pathogen survival (Franz et al.,2008).

The soil microbial community structure also strongly affects soil structure (Young and Crawford, 2004). This, in particular, affects functioning of soils as a powerful water filter and the capacity of this filter with respect to contaminants in both 'green' and 'blue' waters.



Also, soils remain an indispensable source of new powerful antibiotics able to counter the antibiotic resistance dilemma (Ling et al., 2015) and potent medicines to treat such tough-to-treat Diseases as tuberculosis and cancer (Hartkoorn et al., 2012, Liu et al, 2002) Some links between soil and human health tie exposure to soils to immune maturation and, in particular, asthma prevention (von Hertzen and Haahtela, 2006; Rook, 2013) and to mental well-being (Lowry et al., 2007).

To evaluate effects of soil services to public health, upscaling procedures are needed for relating the fine-scale mechanistic knowledge to available coarse-scale information on soil properties and management as health factors. In this context, remarkable advances of medical geology resulted in identification of regions where soils contain components harmful for human health (Selinus, 2013).These results have to be downscaled to evaluate local risks. More needs to be learned about health effects of soils in organic agriculture that are often used for soil quality comparison and benchmarking. The influence of soil degradation and rehabilitation on public health has to be assessed in quantitative terms (Zubkova et al., 2013). Current definitions of healthy soil broadly include aspects that are conducive for human health, and functional evaluation of soil quality with a focus on public health will have useful applications in public policies and perception. The data on soil-health relationships are scarce and very much disjointed, and a concerted international effort appears to be needed to encompass various economic and geographical settings (Brevik and Burgess, 2012) The 'soil-health' connection is complex in character, global in manifestation, and applicable to every human being.


**Essay 3: Water Security / Resources (SDGs 3,6): Soil water and sustainable development goals**

Protecting and enhancing the ability of the earth's soils to provide clean water in sufficient quantities for humanity, ecosystems and agriculture will be a key element in delivering the United Nations Sustainable Development Goals. Soils cover almost all of the ice free terrestrial land surface, making them key for storing and transmitting water to plants, the atmosphere, groundwater, lakes and rivers. It is estimated that 74% of all freshwater appropriated by humans comes from the soil (Hoekstra and Mekonnen; 2012). Not only is it important that soils store and supply water, they also filter it too. Soils are bioreactors. They contain charged surfaces at which exchange reactions can occur; bacteria, fungi and soil animals that process nutrients and contaminants; and act as a media to support plant growth that cycles nutrients and water through the ecosystem. The UN SDG 6 challenges the world to ensure availability and sustainable management of water and sanitation for all. This will not be achieved without protecting and enhancing the ability of the soil to deliver clean fresh water.

Safe affordable drinking water (SDG6.1) will rely on water sources that are reliable and un-contaminated. For 2010 it was estimated that as much as 60% (Baum et al., 2013) of the world's population is not connected to municipal sewage treatment systems suggesting that the remaining 40% of waste water receives no treatment. SDG 6.3 targets halving the proportion of untreated wastewater by 2030. In rural areas this will likely take the form of installing variants of septic systems, which rely on the soil for decontaminating wastewater. It is are also likely that soils will be required to recycle a larger proportion of solid wastes and wastewater (SDG 6.3) from cities and it will be important to understand the capacity of soils to process these inputs and their capacity for assimilating these materials.

The provision of water for crops is of global significance and making the use of this water more efficient (SDG 6.4) is a major challenge. Agriculture amounts to 92% of the globe's freshwater use, far ahead of industrial and domestic usage (Hoekstra and Mekonnen; 2012). Of the 6685 km3/y of water calculated to be used by crops (Siebert and Döll, 2010), it is estimated that 800 to 1100 km3/y is supplied for irrigation from rivers, lakes, reservoirs and groundwater (Döll et al., 2013), as we strive to deliver food security (SDG 2) the volume of water required from these sources is likely to increase. By protecting and enhancing the soil's ability to store and supply water to plants through better soil management there is the potential to make better use of rainwater. Adding just 10 mm per year to plant available soil water across the 306 Mha of irrigated land (Siebert et al., 2015) would provide an additional 30 km3/y of water that could potentially be used by crops and reduce irrigation water requirement.

Soil is the conduit for the vast majority of diffuse pollutants. Nutrients from agricultural sources are responsible for the pollution of lakes, rivers and seas; in many cases bringing about significant degradation of their ecosystems and damaging them as economic and social resources for the people who rely on them for their wellbeing. Restoration of these ecosystems will require restorative actions in the wider catchment, including



better soil management to reduce diffuse pollution (Deasy et al., 2009). However, although soils are excellent buffers against diffuse pollution, they are also slow to change. Therefore, if water related ecosystems are to be restored by 2030 in line with SDG 6.6 significant actions will need to occur urgently.

Managing soils for a better water environment cannot occur without the support and efforts of local communities, many of who fully understand the inexorable link between soils and water, their efforts need to be supported and strengthened (SDG 6.8).


**Essay 4: Climate Change (SDG 13): Impact of climate change on soils and opportunities for mitigation.**

Predicting the response of soils to climate change is extremely important as the top metre of soils globally contain 3 times as much carbon as the atmosphere (Smith, 2004). Small changes in soil carbon stocks can therefore have important impacts on climate – if soil carbon is lost, it could provide a positive feedback to climate warming (Cox et al., 2000). On the other hand, if soils can be managed to store more carbon, they can help to reduce the amount of carbon in the atmosphere, and thereby mitigate climate change (Lal, 2004). This is the aim of the recent proposal at the COP 21 of UNFCCC by the French Government for a global initiative (http://agriculture.gouv.fr/sites/minagri/files/4pour1000-gb_nov2015.pdf) for achieving a "4‰" annual growth rate of the soil carbon stock that would make it possible to stop the present increase in atmospheric $CO_2$.

Climate change has complex impacts on soils. Increasing temperatures will tend to increase decomposition, but this will be limited where soils become very dry – so changes in temperature and precipitation can have additive effects, or may work in opposite directions. In addition, increasing temperatures can also increase plant production, thereby increasing carbon inputs to the soil. This may also decrease the direct impact of climate change on soils and may increase soil carbon (Smith, 2012). Changes in precipitation patterns and amounts will also influence soil organic carbon stocks through their effect on dissolved organic matter production and mobility (e.g. Jansen et al., 2014). This not only affects the soil carbon stock itself, but also couples it to the carbon cycle in aquatic systems (Jansen et al., 2014). While climate change clearly affects soil organic carbon stocks, the magnitude of the effect depends on the intricate interplay of local external factors, such as climate, and the ecosystem specific composition of the organic matter itself that steers its interactions with the inorganic soil phase (Schmidt et al. 2011). As a result not only do soil organic carbon stocks vary vastly between ecosystems, but so does their predicted response to climate change (e.g. Tonneijck et al., 2010).

Nevertheless, while modelling studies (Gottschalk et al., 2012) confirm there is considerable regional variation, with some regions gaining in carbon and some regions losing carbon, globally, climate change is projected to increase soil carbon stocks on mineral soils (i.e. non-peaty soils). On the other hand, peatlands, which contain enormous stocks of carbon (similar to the quantity of all carbon in the atmosphere), may be more susceptible to climate change. When these soils heat up, or if they become drier, vast quantities of carbon could be lost. Similarly, permafrost soils may lose carbon when they thaw (Joosten et al., 2015).

Given the complex interactions between temperature and moisture, between increased productivity and increased decomposition, and variations between regions and different types of soil, predicting the composite effects of climate change on soils is extremely difficult (Smith et al., 2008a).

As well as soils being affected by climate change, improvements in soil management can be used to reduce greenhouse gas emissions or increase soil carbon stocks (Lal, 2004; Smith, 2012). Soil management can therefore be used as a climate mitigation option (e.g. Tonneijck et al. 2010). This is important for climate mitigation, and also to meet UN Sustainable Development Goals (SDG), since SDG 13 is to "Take urgent action to combat climate change and its impacts".

Results from a recent global analysis of greenhouse gas mitigation options in agriculture (Smith et al., 2008b) show that there is significant potential for soils to mitigate GHG emissions, but that the realisation of this potential will depend on the price of carbon. The maximum technical mitigation potential from soil carbon sequestration is around 1 Gt (thousand million tonnes) of carbon per year, but the economic potential at carbon prices between 20 and 100 US$ per tonne of $CO_2$-equvalents is 0.4-0.7 Gt carbon per year (Smith et al., 2008b; Smith, 2012). This means that soil carbon sequestration could be an important part of future climate mitigation portfolios.


**Essay 5: Biodiversity (SDG 15): Functions of soil biodiversity**

Sustainable Development Goal (SDG) 15 aims to 'sustainably manage forests, combat desertification, halt and



reverse land degradation, and halt biodiversity loss'. SDG 15 recognizes that soil micro-organisms and invertebrates are key to ecosystem services, but highlights that their contributions are poorly known and acknowledged. A large fraction of the Earths' biodiversity can be found underground. One square meter of land may easily contain some 5,000-10,000 'species' of viruses, bacteria, fungi, protozoa, nematodes, Enchytraeids, Collembola, mites, earthworms, insects, and some vertebrates. There is mounting evidence that this soil biodiversity contributes to biogeochemical cycles, aboveground biodiversity, soil formation, the control of plant, animal, and human pests and diseases, and climate regulation. Soil biodiversity also contributes to ecological-evolutionary dynamics in ecosystems, which is important for mitigation and adaptation to human-induced global changes in climate, land use, and species gain and loss (Bardgett and van der Putten 2014).

Although much is still to be learned about the distribution of soil biodiversity across the globe, it is becoming evident that it is negatively affected by many human activities, including land use change and management intensification. The first global assessment of soil biodiversity has been completed by the Global Soil Biodiversity Initiative (GSBI) and will be presented as the Global Atlas of Soil Biodiversity, due to be released early 2016 (https://globalsoilbiodiversity.org/?q=node/271). Studies at a continental scale have shown that land use intensification universally reduces the species diversity, especially of the larger sized soil organisms (Tsiafouli et al., 2015), which may negatively impact multiple ecosystem functions and services (Wagg et al., 2014), and their resistance and resilience to extreme drought, leading to enhanced carbon and nitrogen loss to the drainage and ground water during subsequent rainfall events (de Vries et al., 2012). Land use intensification, therefore, may result in loss of ecosystem stability with negative consequences for the Earths' atmospheric composition and water quality.

Loss of soil biodiversity might also result in decreased control of plant, animal, and human diseases (Wall et al. 2015), modify vegetation dynamics, and impact soil physical properties, with consequences for ecosystem services related to soil formation and water regulation (Six et al., 2002). There is evidence that soil biodiversity is also susceptible to invasions and extinctions, nitrogen enrichment (Treseder 2008) soil sealing (Gardi et al., 2013), and climate change (Blankinship et al., 2011). Also, predicted increases in soil erosion and climate-induced shifts in land use, pose a considerable threat to soil biodiversity; however, in all these cases, the full magnitude still needs to be established. Moreover, there are several complications in doing so, including our limited knowledge on what biodiversity is actually present in soils, and its enormous variation in spatial distribution from micro to macroscale (Ettema and Wardle 2002). Many factors have been identified as determinants of soil biodiversity patterns, including pH, soil structure, soil organic matter, and plant diversity and composition, but the relative contributions of each of these factors is still largely unknown. Measures that may promote soil biodiversity include reduced soil tillage, increasing soil organic matter, erosion control, prevention of soil sealing and surface mining activities, and prevention of extreme soil perturbation.


**Essay 6: Land Management (SDG 2, 13, 15): The challenge to implement effective soil conservation.**

Sustainable development goal 15 focusses on sustainable use of terrestrial ecosystems, combat desertification and halt and reverse land degradation. Many ecosystem services and soil functions (Table 1) are connected to this SDG. To reach the desired sustainable situation, good land management plays an essential role. To illustrate the way ahead for in land management, the fragile ecosystems of the Mediterranean are taken as an illustration. When looking back in time, the Mediterranean landscape was managed in a sustainable way for millennia. This changed the landscape (e.g. terraces) and ecosystems (e.g. extensive irrigation systems) to a man-made system (Boogaard, 2005, Stanchi et al., 2012). However, over the last 30 years the land management strategies changed due to altered socio-economic conditions. These changes transferred this sustainable system to be pushed towards, and sometimes over, certain thresholds that caused the system to collapse (Lesschen et al., 2008, Arnaes et al., 2011). To illustrate, we can observe since the 1960's, two contradictory trajectories in the management of soil developments. On the one hand part of the traditionally fully agronomy oriented society has been altered resulting in abandoned ghost towns and whole regions that lost most of the population and were abandoned (Lansata et al., 2005). Former fields and terraces are now overgrown and shrubs and sometimes a full forest have developed. This compromised many of the ecosystem services as listed in table 1 and in addition causes a threat to society due to an increase in the risk of wild fires resulting from the abundant fuel in the new forests. To reach a sustainable situation as described in SDG 15, there is an urgent need to reduce the large wildfires by re-introducing extensive forms of agriculture and grazing in the Mediterranean mountains, thereby reducing the risk of fires and the environmental problems they trigger: soil erosion, water pollution and changes in landscapes and soil properties (Cerdà and Lansata, 2005).





The other trend that can be observed in many countries around the Mediterranean is agricultural intensification. Small scale, sustainable orchards are removed to make room for large scale orchards that are under drip irrigation that contains all nutrients for the plants, making the soil no longer a needed resource for the land owner (Cerdà et al., 2009). Intensification of industrialized agriculture may lead to excessive application of agrochemical leading to pollution of ground- and surface waters and to erosion when lower organic matter contents result in a quality decrease of soil structure. This kind of agriculture may be economically attractive, while the traditional farming systems are no longer economically viable, the sustainability of these new systems is bringing us further away from reaching the objectives of SDG 15. In addition, farmers cling to habits such as keeping their soil 'clean', without weeds; erosion prevention measures such as mulching and cover crops are seen as sloppy management, even though these kind of practices are known to aggravate soil erosion (eg. Keesstra et al., 2009; Cerdà et al., 2009).

Soil management in the Mediterranean type ecosystems needs a new generation of managers, farmers, policy-makers, and also scientists that will understand the importance of the soil system. For this, education programs are needed, starting at the primary school level. Educating the people to acknowledge the importance of soil for soil functions and in the end ecosystem services important for all, may lead to the promotion of organic farming, mulching and minimum or zero tillage. But also the opinion of the consumers, the public can have a strong impact. The public should be aware of the possibility of chose products of higher quality while environmental pollution with agrochemicals is strongly reduced.

Footnote: essay 1 was contributed by Pablo Tittonell; essay 2 by Yakov Pachepsky, essay 3 by John Quinton, essay 4 by Pete
Smith and Boris Jansen, essay 5 by Wim van der Putten and Richard Bardgett and essay 6 by Artemi Cerdà and Saskia
Keesstra.
**4. Actions to be taken**
The six short essays above illustrate the role that soils play when studying major environmental
issues, many of which related to SDGs, as indicated (Table 1 and 2). Clearly, more cooperation of soil
scientists with agronomists, hydrologists, climatologists, ecologists, social scientists and economists
(see also Fig. 1) in interdisciplinary research is desirable to derive meaningful contributions to general
ecosystem services, and recommendations to this effect have been made before and are therefore
hardly enlightening anymore. Here, we would like to emphasize two other issues that we think are
crucial for future activities in soil science. The first issue is the need for a systems approach, where
soil science provides leadership as the environmental issues discussed are interconnected and land-
related, and the relevant processes interact in the pedosphere. The second issue is that the potential
of soils to contribute to solving the major societal challenges of our time, represented by the SDGs,
can only be obtained if we succeed to raise awareness of the crucial importance of soils in supporting
life and livelihoods. Such awareness should register more clearly with the general public,
stakeholders, business leaders and policy makers.

**4.1 The need for a systems approach**
Ecosystems are characterized by interacting geological, hydrological, climatological, ecological and
anthropogenic processes. Due to strong interactions between these processes, a systems approach is
needed to understand the response to changing circumstances in any of the individual elements;



feedbacks within the system may result in unexpected and/or delayed responses to changes.
Approaches will have to reach across levels of integration: in biological terms from species,
communities to ecosystems as has been achieved in ecosystem studies linking below ground
activities to above ground plant development (e.g. Bardgett and Wardle, 2010). In soils, pedon
studies are scaled up to catena's, watersheds, regions and beyond. Food security, for example, is
strongly affected by available nutrients and water resources, climate change, land management and
biodiversity preservation that have different effects at different spatial and temporal scales, and the
same is true for each of the separate issues in relation the all the others. The type of land use
determines these interacting processes and as soils are a key element in determining land use, they
provide a solid foundation for a systems approach. Soil scientists are in a unique position to act in
this capacity. Their history includes extensive interaction with stakeholders when, for example,
developing fertilization practices, preparing soil surveys and combatting land degradation (e.g.
Adimassu et al., 2014; Musinguzi et al., 2015).
At this point in time the question can be raised as to who will seize the initiative to start such broad
inter- and transdisciplinary studies, focusing on ecosystems but with a clear soil component.
(interdisciplinarity refers to disciplines working together; transdisciplinarity also involves
stakeholders). Funding agencies such as the EU HORIZON 2020 scheme and its predecessors have
clear ambitions to realize this type of research approach and many ecological and climatological
system studies have been made, particularly for larger regions. But integrating climatological,
hydrological, agronomic and ecological aspects is more difficult, certainly when including socio-
economic aspects. The six major environmental issues, covered in the six essays relating to SDGs
presented above are land-related, and soil scientists are therefore in a natural, but also highly
challenging, position, to initiate, guide and complete systems analyses of ecosystems, working with
fellow scientists, stakeholders and policy makers. This applies at different spatial scales, ranging from
fields, farms and regions to the world at large. It also applies at different temporal scales, ranging
from present day processes, to geological times to understand system responses and feedbacks.
Such integrated studies are still relatively rare, thus presenting a new research "niche". An example is
a comprehensive, integrative study of innovative dairy systems in the Netherlands using Life Cycle
Assessment to characterize the entire production chain, including an economic and energy analysis.
Improvement of nutrient cycling resulted in improved groundwater quality, lower emissions of
greenhouse gasses and lower energy use, higher organic matter contents of the soils and incomes,
the latter due to lower costs. Biodiversity was high because of preservation of hedgerows along
relatively small fields. Dolman et al. (2014) presented results at farm level and de Vries et al. (2015)





scaled the work up to a regional level. Van Grinsven et al. (2015) extended the work to a broad policy
analysis, considering future development scenarios.
In the end, effective communication of results to citizens, stakeholders and policy makers is crucial
and the example of the UNFCCC, that defines "lighthouses" for successful case studies, is
inspirational in this context.

**4.2 Creating and sustaining awareness**
Awareness raising by establishing genuine two-way dialogues, requires different approaches when
addressing policy makers, stakeholders, the public and colleagues in other disciplines even though a
common theme will emerge at the end of this section. To improve the connection with policy makers
it is important to consider their way of reasoning and two approaches may be helpful in this context,
following: (i) the policy cycle when planning and executing research, which includes signalling and
definition of a given problem taking into account the opinions of all involved, design, decision,
implementation and evaluation (e.g. Althaus et al., 2007, Bouma et al., 2007). Many current research
projects spend most of their time on design and relatively little time on signalling which may lead to
hastily conceived plans and disengagement of stakeholders who feel left out. Also, implementation is
often seen as the responsibility of others while it is crucial to demonstrate – if successful - the
relevance of soil science in the design and implementation of such projects (e.g. Bouma et al., 2011).
Nothing is as convincing as a successful project! (ii) the DPSIR approach (Skondras and Karavitis,
2015) can be useful when performing land-related research, it distinguishes external drivers,
pressures, impacts and responses to land-use change that affect the state of the land in past, present
and future (e.g. van Camp, 2008; Bouma et al., 2008; Mol and Keesstra, 2012).
So rather than jumping right away into agronomic, hydrological, climatological and ecological studies,
or even into a comprehensive systems analysis, signal the current land-use drivers, the pressures
they generate and the impact they have. Doing so, it pays to involve stakeholders and policy makers
at an early moment in a "joint-learning" mode; also referred to as co-production of knowledge. This
includes characterization of actual as well as a range of possible future conditions as a source for
decisions to be taken. In close interaction with all stakeholders involved, design possible alternatives
and explore ways to have one of them approved and implemented. The design phase involves major
input by research, acknowledging that much information and knowledge is already available as is
clearly demonstrated in the first six essays. New research can be based on observed gaps during the
signalling and design process.



Stakeholders have a direct personal or commercial interest in the way land-use issues are
investigated. SDGs have a societal focus and future soil science research can only be successful if
stakeholders are part of the research effort in transdisciplinary projects, based on the principle of
time-consuming "joint-learning" which is facilitated by providing accessible narratives about case
studies (Thomson Klein et al., 2001; Bouma et al., 2015; Bouma, 2015b). The increasing importance
of transdisciplinarity also implies that the "top-down, command-and-control" character of much
current environmental legislation should evolve into a "bottom-up, joint learning" mode that truly
engages modern stakeholders and is an important ingredient of adaptive management (e.g. In't Veld,
2010). One additional interesting tool to involve stakeholders and the general public are projects
using citizen science (Bonney et al., 2014). The further development of such projects and the
development of voluntary soil governance instruments is the way forward for such innovative
bottom-up participatory approaches. Strengthening voluntary partnerships, like the Global Soil
partnership (GSP) could ultimately lead to a more effective sustainable soil management then many
of the, largely not implemented, mandatory legal frameworks (Montanarella, 2015).But awareness is
hampered by the gradual and slow character of changes in the pedosphere. Even abrupt changes in
driving forces (e.g. climate, land management) will result in slow changes in soil properties, and often
delayed response in the quality of soil ecosystem services. Such gradual and delayed behaviour does
not attract the kind of attention reserved for natural hazards like volcanic eruptions, earthquakes,
tsunami's and floods. Yet the consequences of soil degradation for society as a whole will be more
severe than any of those (local) phenomena. Another issue is that with the green revolution, the
connection of food and soil has lost visibility and importance (essay 6). Not only are city dwellers less
aware of where their food in the supermarket originates from, even some farmers consider their
land as an industrial production factor that can be manipulated at will, ignoring ecological thresholds.
Essay 1 articulates relevant approaches for resource-poor small–scale farmers in developing
countries. But questions have been raised whether or not high food demands of mega-cities in future
will require a significant productivity increase of land and labor that is associated with more large-
scale farming (e.g. De Ponti et al, 2012). That new and effective antibiotics are being derived from
soil and that human health can be negatively affected by soil-borne diseases, as described in essay 2,
is unknown to the public. The international One Health initiative
(http://www.onehealthinitiative.com) focuses on links between human and veterinary medicine and
environmental science but pays so far little attention to soils. The public at large does not recognize
either the crucial and fundamental importance of biodiversity to life on earth, as discussed in essay 5.
That the quality of ground- and surface water is, to a large extent, governed by percolation through
soil or by surface runoff that may result from soil compaction or surface sealing (Essay 3) is unknown
as well. That there is more organic matter in soils than in all the tropical forests combined and that



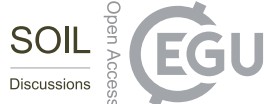

carbon sinks in soil present a major mitigation opportunity (as described in essay 4) has drawn
considerably less attention than reducing $CO_2$ emissions. So proper communication of the role of
soils, applying modern communication practices, is urgently needed, taking a positive approach and
emphasizing successful examples and programs. Complaining that soils have not received the
attention they deserve serves no useful purpose.
Creating awareness with colleague scientists presents an intriguing dimension to this discussion. The
need for interdisciplinarity has been discussed above. But how can interdisciplinarirty be realized?
Scientists of a given discipline are only accepted as partners in interdisciplinary projects if they can
deliver input that is considered to be of substantial added value by the other partners. Many
agronomists, hydrologists, climatologists, ecologists, let alone economists and sociologists, are not
aware of what soil scientists have to offer. A recent example on: "Climate-smart agriculture" by
Bonfante and Bouma (2015) illustrates this point. By running a crop production simulation model,
considering the effects of climate change, growing eleven maize hybrids and different degrees of
irrigation water availability for a Mediterranean area, they showed that agronomic and irrigation
plans had significantly different effects on different soil types occurring in the area. These results
allowed rational future planning of cropping and irrigation schemes, and were welcomed by farmers
and irrigation engineers, who were rather surprised to see these soil-based results. An example for
developing countries demonstrated within-farm nutrient gradients which strongly affected yield
response requiring alternative location-specific approaches in contrast to the traditional blanket
application of fertilizers (Tittonell et al., 2008). Again, documentation of soil differences had a
significant effect on management. Of course, there are more of such examples and they should be
presented more prominently.
The example of the UNFCCC, producing "lighthouses" for successful programs, is inspiring in this
context because presenting soil-based "lighthouses" is the overall connecting theme for awareness
raising. The good news is that many "lighthouse" examples are there, but we have not yet recognized
the urgency to communicate these examples in an effective manner, also showing what might have
happened without soil science input. Modern communication is a science, or better, an art, that
cannot be accomplished solely as a side activity by scientists who were trained in entirely different
fields. Many of our current scientific journals are not focused on publishing "lighthouse" papers and
finding appropriate outlets for this work is still a challenge (e.g. Bouma, 2015a). As for the MDGs,
there is the need to demonstrate that the SDGs can be implemented successfully at local level. As the
Millennium Villages Project (Sanchez et al., 2007) has been demonstrating for the MDGs, there is the
need for a similar project for the SDGs in the future.



### 4.3 How to overcome constraints

To be realistic, several constraints have to be recognized when proposing a central role of soil scientists in initiating and guiding inter- and trans-disciplinary projects, aimed at land-related aspects of the SDGs. Constraints when raising awareness have already been discussed above, but social and economic constraints as well as policy barriers require additional attention.

The first level of constraint is a social. As we learn from essay 6, a good farmer in Spain is considered to be a farmer that keeps his or her fields tidy and clean, apparently unaware of the resulting vulnerability to erosion in sloping areas. A farmer that leaves weeds on the field is considered to be a sloppy farmer by peers. Even though there is a wealth of information on successful forms of soil management that leads to less erosion and degradation (e.g. WOCAT, 2007, Schwilz et al., 2012; Cerdà et al., in press) implementation in practice is delayed, often for social reasons. Intensive agricultural practices that are accepted by commercial farms may lead to environmental pollution by biocides and excess fertilizers (Roy and Mcdonald, 2013; Shi et al., 2015; Sacristàn et al., 2015). The language and perceptions of farmers and environmentalists are still quite different, even though mutual understanding has increased in many countries. In developing countries, the situation is often even more difficult because of population growth, increasing the pressure on land and water resources. Land vulnerable to degradation is taken into cultivation with adverse effects on the soil functions and ecosystem services (Fialho et al., 2014; Olang et al., 2014; Costa et al., 2015). Competing claims on land by industry, urban sprawl, agriculture and nature are all too often not decided by rational arguments but by political or ideological arguments. To disrupt this negative discourse and provide a counterweight to negative social pressures, education is important and so are specific examples of successful management systems. But most convincing may be a demonstration that good environmental practices can correspond with positive economic effects: "what is good for the environment can be good for business" (see also essay 1) - after all "money talks". Fine-tuning application of agrochemicals to the needs of the plants can, for example, strongly reduce costs for the farmer, increasing net income while soil quality is improved (e.g. Dolman et al., 2014; De Vries et al., 2015); and reduce the pressure on the natural ecosystem. Many positive examples are there to be shown and this deserves more attention in future. Intercropping, strip-cropping or the use of mulch can result in higher yields, stronger resilience and larger biodiversity (Whitmore and Schroeder, 2007; Novara et al., 2013; Laudicina et al., 2015). With appropriate land management, intensified farming may result in higher production combined with increased soil organic matter content (Govers, this issue).

The second level of constraint is economic. Farmers everywhere have to make a living and economic results of any commercial farming operation must be positive to be sustainable from a livelihood

 

point of view. Here, the previous point applies as well. Demonstrating with quantitative procedures
that striving for sustainable development does not necessarily imply loss of income, but may increase
incomes in the short, medium or long term, is crucial because in the information age words by
themselves will not convince anyone. Including an economist in the team allowed important
conclusions as to farmers income in a systems analysis of dairy systems in the Netherlands (Dolman
et al., 2014). Specific examples are needed, also considering the important issue of land ownership
and tenure. Land owners are traditionally more inclined to invest in their property while tenants are
more focused on short term benefits (Teshome et al., 2015, Marques et al., 2015). But environment
friendly practices may pay off even in the short run, and this will also be convincing for tenants. The
simple and obvious statement that: " land" has a price, while "soil" has not, has major implications
when debating soil contributions to sustainable  development because items that cannot be
expressed in monetary terms tend to lose attention when, as so often, financial aspects dominate
the debate.
The third and last level of constraint is the policy barriers. Politicians in democratic systems in the
information age tend to be risk-averse and focused on activities that can generate favourable media
exposure to their voters in the short term (Bouma and Montanarella, this issue). They are constantly
approached by lobbyists and choosing potential "winners" appears to become ever more important.
So far, soil issues do not play a significant role in such strategic deliberations. Major policy changes all
too often result from disasters and a major problem for soil science is the fact that soil degradation is
a creeping phenomenon that does not attract media attention. Of course, mudflows and flooding are
often associated with poor soil management in upslope watersheds, but this link is not always well
communicated. In general, policy aspects manifest themselves at three levels: strategic, tactical and
operational. Providing examples of successful projects, as discussed above, can help to enable
politicians to make sustainable decisions, but the effect is bound to be limited as ideological
standpoints do not need to rely on evidence. Still, it is important to at least try to speak the language
of the policy arena. That is why attention was paid in discussions above to the policy cycle and to the
DPSIR procedure. More promising in the information age are bottom-up actions of engaged
stakeholders who are the voters that ultimately, at least in democracies, determine the fate of any
politician. Soil scientists would be well advised to connect with NGO's and local initiatives that focus
on sustainable development. Moreover, measures to reduce soil degradation are usually expensive
and do not provide revenues immediately. Legislation for soil protection is therefore unpopular.
Finally, the assessment and monitoring of soil quality is tedious as soil is heterogeneous in nature
and good monitoring methodologies are expensive or even non-existent. Continued attention for
streamlining and developing innovative procedures is therefore needed, and the introduction of



remote and proximal sensors may make important contributions in this context (Viscarra Rossel, et
al., 2010, Stoorvogel et al., 2015).
In conclusion, political barriers are severe but they can be overcome by developing convincing
examples of land-related sustainable development that voters can present and lobby for when
engaging with politicians.

**4.4 Implications for the soil science discipline:**
Soil scientists are becoming aware of their central role in initiating the systems approach necessary
to combine aspects of different disciplines. Although many soil science projects are still highly
disciplinary, examples are increasingly available to demonstrate successful results of inter- and trans-
disciplinary studies. (e.g. Mota et al., 1996; Schröter et al., 2005; Tittonell et al., 2010; Dolman et al.,
2014; de Vries et al., 2015; Berendse et al., 2015, Keesstra et al., 2012, Brevik et al., 2015, Torn et al.,
2015). Such studies advance the knowledge base by including basic research, which is crucial to
maintain a vital scientific discourse and develop novel solutions for societal challenges. Using
methodologies developed and established in other disciplines can solve problems in other fields that
have been lingering for decades.
But within soil science itself, work remains to be done. An example is the comparability of methods
and data. Measured data are usually assumed to represent the truth and are used for calibrating
models and executing scenario analysis for decision making. However, the value of data is
determined by the experimental set up, the sampling scheme and the measurement technique itself.
Too often data are used without considering these constraints. An example is the widespread,
indiscriminate use of pedo-transfer functions. To be able to transfer data from one research project
to the next it is important to validate and harmonize technologies and methodologies, and
standardize information to achieve sound science allowing reliable translation into relevant
information for stakeholders.
The key to establish more effective inter- and trans-disciplinary, holistic research is to communicate
to stakeholders, business leaders and policy makers, to reach out and to invite scientists from other
disciplines to participate. This requires special abilities that are not being taught in current scientific
education. We should educate "knowledge brokers" that have the ability to inject the right type of
knowledge to the right person at the right time and place. One important constraint for new
developments is the way science is funded at this time, stimulating competition rather than
collaboration.





**4.5 Is there a key message from soil science?**
The public needs to become more engaged with soils because changes to sustainable forms of land
use are only possible when children, farmers, citizens, teachers, business leaders and policy makers
become more aware of the central function of soils in our society. This not only calls for relatively
simple messages, but also for symbols and narratives that appeal to people. Greenhouse gasses are a
universally known symbol for climate change and so are polar bears to illustrate warming of the ice
caps. Economists use the Gross National Product (GNP) and particularly its growth % as a well-known
symbol of material well-being that is embraced by the political arena. Pictures of hungry children
illustrate the concept of food security.
For soils, the organic matter content of mineral soils could be a suitable symbol for soil quality as it
positively affects most soil functions. Higher organic matter contents in a given soil increases its
adsorptive capacity for nutrients and water and improves soil structure and its stability. Soil organic
carbon is also associated with a higher biodiversity that is a proper symbol for a "living soil", and last
but not least, increased soil organic carbon stocks will mitigate atmospheric $CO_2$ concentrations. Of
course, this has been known for a long time by soil scientists but identifying a suitable symbol for
soils cannot be based on knowledge alone but needs to be easily accessible and to somehow trigger
the imagination of outsiders. From a practical point of view, soil organic matter contents are
relatively easy to measure, most recently also by handheld proximal sensors allowing real-time
monitoring of changes of soil organic carbon in time and space (e.g. Viscarra Rossel, et al., 2010,
Stoorvogel et al., 2015). Given the possible role of soils in climate mitigation, and their role in
underpinning sustainable development, the lasting legacy of the International Year of Soils in 2015
should be to put soils at the centre of policy supporting environmental protection, sustainable
development, and the delivery of climate mitigation (Smith et al., 2015). An important challenge, and
essential contribution from the scientific community, will be to provide the guidance and expertise
needed to effectuate sustainable carbon sequestration. Given the complex interplay of (local) factors
that govern the carbon sequestration (potential) in the various soils and ecosystems of our planet,
rigorous scientific underpinning is needed to device tailor-made location-specific soil management
schemes aimed at optimizing carbon sequestration whilst acknowledging other important ecosystem
services. In addition, there is a need for cheap and reliable monitoring of (trends in) soil organic
carbon content, preferably by remote sensing.

**5. Recommendations**





- **Embrace the SDGs.** The UN Sustainable Development Goals provide a widely recognized societal framework that allows soil science to demonstrate its relevance for realizing a sustainable society by 2030.

- **Show the specific value of soil science:** Research should explicitly show how using modern soil information can improve the results of inter- and trans-disciplinary studies on SDGs related to food security, water scarcity, climate change, biodiversity loss and health threats. Implications for society should be communicated in terms that appeal to stakeholders, citizen at large and the policy arena. Well documented and specific examples ("lighthouses") are most effective.

- **Take leadership in overarching systems-analyses of ecosystems:** Given the integrative nature of soils, soil scientists are in a unique position to initiate and guide a comprehensive systems analysis of ecosystems, integrating land-related SDGs.

- **Raise awareness of soil organic matter as a key attribute of soils** to illustrate its importance for soil functions and ecosystem services. Show how soil management can manipulate the organic matter content and quality of any given soil.

- **Improve the transfer of knowledge.** Inter- and trans-disciplinarity requires effective communication of soil knowledge and expertise to outsiders with little knowledge about soils. Knowledge brokers with a soil background can play an important role here. They should be professionally selected and educated.

- **Start at the basis:** Global citizens have access to an ever-increasing volume of data on the internet, some of it relevant, much of it of dubious quality. As educational standards increase, global citizens will use this information to form opinions and make decisions. Our task is to insert our evidence-based knowledge in the opinion-forming and decision-making process at the right time and place, and in the right way. This fits well within the citizen-science concept. Overall, educational programs are needed at all levels, starting in primary schools, and emphasizing practical, down-to-earth examples.

- **Facilitate communication with the policy arena:** frame research in terms that resonate with politicians in terms of the policy cycle or by considering drivers, pressures and responses affecting impacts of land use change. Approaching the policy arena through stakeholders and citizens may, however, be most effective in the information age.

- **Collaborate beyond the comfort zone:** All this is only possible if researchers look over the hedge towards other disciplines, to the world-at-large and to the policy arena, reaching over to listen first, as a basis for genuine collaboration.

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










Table 1: The UN "Sustainable Development Goals" for the period 2015–2030. ((http://sustainabledevelopment.un.org/focussdgs.html), related to ecosystem services and soil functions, as discussed.).

| SDGs topic | | Provision of food, wood and fibre. (1) | Provision of raw materials. (2) | Provision of support for human infrastructures and (3) | Flood mitigation (4) | Filtering of nutrients and contaminants (5) | Carbon storage and greenhouse gases regulation (6) | Detoxification and the recycling of wastes (7) | Regulation of pests and disease populations (8) | Recreation (9) | Aesthetics (10) | Heritage values (11) | Cultural identity (12) | Relates to soil function (Table 2) |
|---|---|---|---|---|---|---|---|---|---|---|---|---|---|---|
| 1 | End poverty in all its forms everywhere | X | X | X | X | | | | | | | | | 1, 5 |
| 2 | End hunger, achieve food security and improved nutrition and promote sustainable agriculture | X | | X | | | | | | | | | | 1, 2, 4 |
| 3 | Ensure healthy lives and promote well-being for all at all ages | X | | | | | | | X | X | X | X | X | 1, 2, 3, 4, 5, 7 |
| 4 | Ensure inclusive and equitable quality education and promote lifelong learning opportunities for all | | | | | | | | | | | | X | 7 |
| 5 | Achieve gender equality and empower all women and girls | | | | | | | | | | | | | |
| 6 | Ensure availability and sustainable management of water and sanitation for all | | | | X | X | | X | | X | | | | 2 |
| 7 | Ensure access to affordable, reliable, sustainable and modern energy for all | X | X | | | | | | | | | | | 1, 5, 6 |
| 8 | Promote sustained, inclusive and sustainable economic growth, full and productive employment and decent work for all | X | X | X | | | | | | | | | | 1, 2, 5, 6 |
| 9 | Build resilient infrastructure, promote inclusive and sustainable industrialization and foster innovation | | X | X | | | | | | | | | | 2, 4, 5 |
| 10 | Reduce inequality within and among countries | | | | | | | | | | | | | |
| 11 | Make cities and human settlements inclusive, safe, resilient and sustainable | | X | X | | | | | | | | | | 2, 4, 5, |
| 12 | Ensure sustainable consumption and production patterns | X | X | | | X | X | X | | | | | | 1, 2 |
| 13 | Take urgent action to combat climate change and its impacts | | | | X | | X | | | | | | | 2, 6 |
| 14 | Conserve and sustainably use the oceans, seas and marine resources for sustainable development | | | | | | | | | | | | | |
| 15 | Protect, restore and promote sustainable use of terrestrial ecosystems, sustainably manage forests, combat desertification, and halt and reverse land degradation and halt biodiversity loss | X | X | X | X | X | X | X | X | X | | X | X | 1, 2. 3, 4, 5, 6 |
| 16 | Promote peaceful and inclusive societies for sustainable development, provide access to justice for all and build effective, accountable and inclusive institutions at all levels | | | X | | | | | | | X | X | X | 4, 7 |
| 17 | Strengthen the means of implementation and revitalize the global partnership for sustainable development | | | | | | | | | | | | | |




**Table 2: The seven soil Functions (SFs) as defined by the European Commission (EC, 2006)**

| | |
|---|---|
| 1 | Biomass production, including agriculture and forestry |
| 2 | Storing, filtering and transforming nutrients, substances and water |
| 3 | Biodiversity pool, such as habitats, species and genes |
| 4 | Physical and cultural environment for humans and human activities |
| 5 | Source of raw material |
| 6 | Acting as carbon pool |
| 7 | Archive of geological and archaeological heritage |




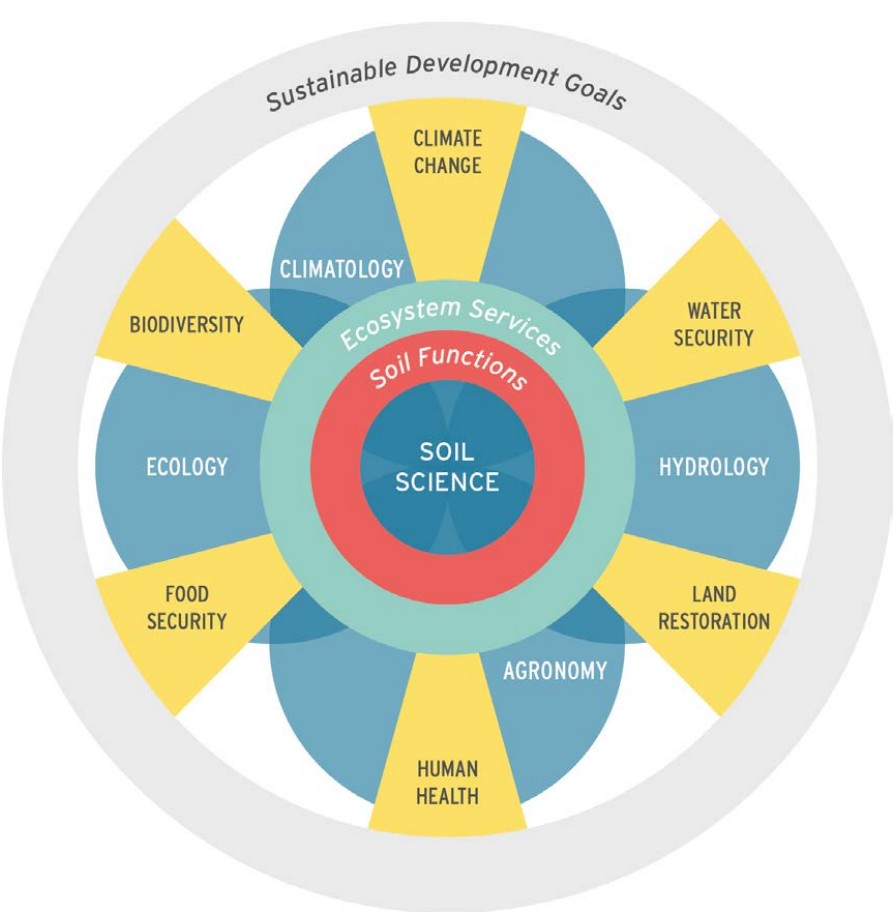


Figure 1 shows six major global issues, each of which relates to one or more of the SDGs: (i) food
security; (ii) human health; (iii) land management, including land restoration; (iv) water security; (v)
climate change, and (vi) biodiversity preservation. Each of these issues will be discussed in short
essays, loosely based on discussions held at the EGU Soil Conference in Vienna, in April 2015 and at
the Wageningen Conference on: "Soil Science in a Changing World" in August 2015.