# Peer review of "The significance of soils and soil science towards realization of the UN sustainable development goals"

_SOIL, 2015_

## Referee Comment (RC1) · Anonymous Referee #1 · 4 Feb 2016

The forum paper tackles the inevitable role of soil science community in achieving the Sustainable Developemnt Goals (SDGs) and as a such discuss the possible directions of further development of the soil science agenda in the coming years. The paper demonstartes the relevance of soil science for SDGs trough the links between ecosystem services and soil functions connected to trans-disciplinary SDGs. It provides insights in the six main issues to which the soil science could have crucial contribution and discuss the action to be taken. The paper is of a great relevance for the soil science community and achieving the SDGs and I fully support its publication. I do have only few remarks, that might improve the readability and presentation.

1. the six key issues are described in six short essays. Each of them has its own

style and way of presentation, what makes reading of section 3 attractive. On the other hand, the direct and quick comparison between the essyas outcomes is difficult. A summary table, which would include: (example for first essay) 1. key issues -identified by experts (role of soils for food security) 2. related SDGs (1,2,3) 3. main challenges (necessity of fast growth of food production vs. smallhorder farming,climate 4. change impact, soil degradation, biodiversity conservation) 5. opprotunities for change rooted in soil science: (restoring soil poductivity and ecosystem function) 6. way to move forward soil science can address: (inovative "precision agriculture", system approach to nutrient acquisition and management, agroecological strategies, soil food web and nutrient and water use efficiency).

This would improve the link to sections 4.1, 4.2. Other possibility would be to divide each essay to subsections concerning the bullet points above. Ideally a subsection / collumn "role of soil science" would be added.

2. Section 4: Actions to take.

According to the title, the readers might seek the answers to questions "which action to take to tackle the SDGs". The interdisciplinary research, system approach managed from soil scientist in the centre, and awarnes raising are named together with many constraints. The complexity of the problem is sufficiently and effectively described (in all three subchapters), but might sound overhelming for soil scientist as it requires change in paradigma within and without of the soil science community. The action to be taken within and without the community should be stated and distinguished more clearly. There are positive examples in other disciplines in meeting the political and societal contrains, that might be used to describe the way "how to take the action", which might be the question of the reader. Section 4.5, it would be interesting to state how to proceed with the outlined agenda beyond the International Year of Soil 2015, in the Interanational Year of Pulses, which have also connection to soils.

Technical comments: p.10, line 177-181, there is no social science and economics in

figure 1 Graphical abstracts address the issues discussed in article much better than Figure 1. Could the graphical abstract be used instead of Figure 1 Unclear meaning of small blue and red circles with white numbers in the graphical abstract
[Figure]

---

## Referee Comment (RC2) · D. Arrouays (Referee) · 16 Feb 2016

I like this paper.

I think it is timely and well suited to a FORUM paper.

I have a couple of remarks and suggestions.

Essay 1 (page 5) Most of the references concentrate on soil degradation and negative aspects. I think it wuld be helpful to add more refernces under bullets i) to iv) to illustrate some success stories and show that there are effective solutions for managing soils for food security
[Figure]

Essay 2 page 6. "Soil eutrophication appears to create favourable ccondition for pathogens survival" So, what? What do you propose?

Essay 3: page 6, line 3 of the Essay 3. I do not agree with the sentence "Soils cover almost all of the ice free terrestrial land surface". especially within the framework of SDGs. On the contrary I would insist on the fact that soils are a limited resource in terms of area, and in terms of the aera we can act on. End of page 6 (Box essay 3): "it is are also likely" .remove "are" . Page 6. Section on irrigation. It is said that 800 to 1100 km3/y are used for irrigation. in comparizon, the additional 30km3/y proposed seems nearly negligible. So I think the example should be re-written to make it more convincing.

Essay 4, page 8. last section. there is an i missing at equIvalents

Essay 5, page 8 line 5. I had in mind much larger numbers of 'species' by square meter than these figures of 5,000 to 10,000. please check and include references. end of the section "still largely unknown" there are some recent paper about earthwoms abundance and diversity in Europe, and on soil microbial diversity in France, you may cite them to show recent advances

Essay 6 page 9 line 10 of the box, there is a (ref) missing Last sentence. I'm not a native English speaker, is "possibility of chose products" correct?

lines 348-360. Is there a "policy behaviour scientist" in the list of authors? I would be very careful in writing this.

line 389-390. Do you have a reference for the widespread indiscriminate us of ptf? You should admit too that often there is no other choice than using them.

Lines 413-414. This is mainly true for cultivated soils and grasslands, on the other hand, in many forested parts of the world, organic carbon accumulates because there in no biological activity and increases acidity. The carbon pool we need for biodiversity is a "living" carbon pool, not accumulation of acid O layers.
[Figure]

lines 419-421. I think you are very optimistic about monitoring changes in SOC with sensors. Given the errotr of measurement they have by know, and given the rate of change in SOC, it will take nearly 50 or 100 y to prove a change. Moreover SOC are not so relatively easy to measure and to monitor given their high short-scale spatial variability. Same for lines 368-370, for remote sensors it is even worse because of atmospheric effects, vegetation cover, etc. Same again for lines 430-431. "preferably by remote sensing" this is a very dangerous assumption. Firts it is not yet operational at all, second, if I am a funding agency, I will never give you again one cent for real measurements but tell you, okay, just look at the satellites images, some of them are free !

Recommendation. I'm surprised not to find anything about data collection and data sharing.

Throuhout the text, I'm also surprised that there is not ref to the recent reports delivered by the ITPS, ad even not to paper by Montanarella et al on this and in this journal.

Refs

The paper by Montanarella in Nature is opublished, please add issue and lines numbers.

Overall a nice paper for the FORUM.

---

## Author Comment (AC1) · 18 Mar 2016

Reply to the topical editor and referees of SOIL for the paper: FORUM paper: The significance of soils and soil science towards realization of the UN sustainable development goals. (Keesstra et al) First of all we would like to thank the reviewers for their time and effort to improve our paper and for the compliments they made about our paper. One other issue that need to be solved is that in the first version of this paper one co-author was accidently not taken up in the author list: Boris Janssen (also one of the guest editors of the special issue. I want to ask to take up his name. Secondly, we would like to remove FORUM paper from the title and just have 'The significance of soils and soil science towards realization of the United Nations Sustainable Devel-

opment Goals.' To efficiently answer the comments of the reviewers we have copied the reviews in this document and put our replies to the comments in italics and in blue colour beneath each section. Anonymous Referee #1 The forum paper tackles the inevitable role of soil science community in achieving the Sustainable Development Goals (SDGs) and as such discusses the possible directions of further development of the soil science agenda in the coming years. The paper demonstrates the relevance of soil science for SDGs trough the links between ecosystem services and soil functions connected to trans-disciplinary SDGs. It provides insights in the six main issues to which the soil science could have crucial contribution and discuss the action to be taken. The paper is of a great relevance for the soil science community and for achieving the SDGs and I fully support its publication. I do have only few remarks that might improve the readability and presentation. Answer: many thanks for the compliments.

1. the six key issues are described in six short essays. Each of them has its own style and way of presentation, what makes reading of section 3 attractive. On the other hand, the direct and quick comparison between the essay outcomes is difficult. A summary table, which would include: (example for first essay) 1. key issues –identified by experts (role of soils for food security) 2. related SDGs (1,2,3) 3. main challenges (necessity of fast growth of food production vs. smallholder farming, climate 4. Change impact, soil degradation, biodiversity conservation) 5. opportunities for change rooted in soil science: (restoring soil productivity and ecosystem function) 6. way to move forward soil science can address: (innovative "precision agriculture", system approach to nutrient acquisition and management, agroecological strategies, soil food web and nutrient and water use efficiency). Answer: We invited recognized experts ( identified in a footnote) in the various fields to reflect on the six major issues. This they did and we therefore would prefer to keep the essays as they are and not rephrase them following a specific , standardized format as suggested .

This would improve the link to sections 4.1, 4.2. Other possibility would be to divide each essay to subsections concerning the bullet points above. Ideally a subsection

/column "role of soil science" would be added.

Answer: We feel that we have addressed the role of soil science already extensively in sections 4.4 and 4.5. We see little opportunity to further extend this as we fear that the paper could become repetitious if we would add more sections.

2. Section 4: Actions to take. According to the title, the readers might seek the answers to questions "which action to take to tackle the SDGs". The interdisciplinary research, system approach managed from soil scientist in the centre, and awareness raising are named together with many constraints. The complexity of the problem is sufficiently and effectively described (in all three subchapters), but might sound overwhelming for soil scientist as it requires change in paradigm within and without of the soil science community. Answer: we fully agree that what we are arguing for will indeed need a change in paradigm within and outside the soil science community. So be it. The matter is complex and making it sound too simple will not serve our goal. But, indeed, the paper presents a major challenge.

The action to be taken within and without the community should be stated and distinguished more clearly. Answer: We feel that we did so already in considerable detail in sections 4.4, 4.5 and 4.6 and in the recommendations. As is, we see little possibility to clarify our statements which we consider to be as clear as possible, considering the complexity of issues being discussed.

There are positive examples in other disciplines in meeting the political and societal constraints that might be used to describe the way "how to take the action", which might be the question of the reader. Answer: this is a most useful comment. We added a sentence indicate 'how to take action'.. We refer now to the climate research community that has effectively communicated their science to the public and to policy makers.

Section 4.5, it would be interesting to state how to proceed with the outlined agenda beyond the International Year of Soil 2015, in the International Year of Pulses, which

have also connection to soils. Answer: Thank you very much for this idea, but rather than refer to the somewhat nebulous International Year of Pulses, we refer now to the paper of Montanarella in Nature, discussing how to proceed after the IYS.

Technical comments: p.10, line 177-181, there is no social science and economics in figure 1 Graphical abstracts address the issues discussed in article much better than Figure 1. Could the graphical abstract be used instead of Figure 1 Answer: The graphical abstract is basically a (simplified) merger between Figure 1 and the two tables in the article. We feel that it is easier for the reader to show them separately in the paper. Unclear meaning of small blue and red circles with white numbers in the graphical abstract. Answer: The figure we made as graphical abstract is already a figure with a very density of information, so we feel we should not add anything more. We agree the figure needs some time to absorb, but we think the colours are indicative and, if studied a bit longer, should speaks for themselves.
* * *

---

## Author Comment (AC2) · 18 Mar 2016

Referee D. Arrouays I like this paper. I think it is timely and well suited to a FORUM paper. Answer: :))ïĄŁ

I have a couple of remarks and suggestions. Essay 1 (page 5) Most of the references concentrate on soil degradation and negative aspects. I think it would be helpful to add more references under bullets i) to iv) to illustrate some success stories and show that there are effective solutions for managing soils for food security Answer: bullet points i) to iv) were not intended to describe a gloom picture but rather to point out challenges ahead of soil science as well as possible pathways. We are of course aware of success stories where soil scientists liaised with multiple stakeholders to contribute to restore

degraded land; but the reviewer will surely agree that the spatial extent of such suc-cess stories is rather limited in the tropics and that many of them are associated with externally supported, hence often short-lived initiatives. We added references describ-ing successfully implemented strategies to mitigate soil degradation and improve soil quality throughout the referred bullet points. Thanks you for the useful suggestion.

Essay 2 page 6. "Soil eutrophication appears to create favourable conditions for pathogens survival" So, what? What do you propose? Answer: we added a sentence to explain why this is needed.

Essay 3: page 6, line 3 of the Essay 3. I do not agree with the sentence "Soils cover almost all of the ice free terrestrial land surface". Especially within the framework of SDGs. On the contrary I would insist on the fact that soils are a limited resource in terms of area, and in terms of the area we can act on. Answer: We removed this part of the sentence.

End of page 6 (Box essay 3): "it is are also likely" .remove "are" . Answer: done

Page 6. Section on irrigation. It is said that 800 to 1100 km3/y are used for irrigation. in comparison, the additional 30km3/y proposed seems nearly negligible. So I think the example should be re-written to make it more convincing. Answer: we agree with the reviewer that this does not sound very convincing, so we rephrased the sentence to remove the numbers from the sentence.

Essay 4, page 8. last section. there is an i missing at equIvalents Answer: done

Essay 5, page 8 line 5. I had in mind much larger numbers of 'species' by square meter than these figures of 5,000 to 10,000. Please check and include references. End of the section "still largely unknown" there are some recent paper about earthworms abundance and diversity in Europe, and on soil microbial diversity in France, you may cite them to show recent advances Answer: We have checked this and, indeed, the numbers are much larger. We have adapted the text along these lines

Essay 6 page 9 line 10 of the box, there is a (ref) missing Last sentence. Answer: we are not sure what the reviewer means with this comment.

I'm not a native English speaker, is "possibility of chose products" correct? Answer: Indeed the English was not correct and we adapted the text.

Lines 348-360. Is there a "policy behaviour scientist" in the list of authors? I would be very careful in writing this. Answer: In the author team there are at least two people (Bouma, Montanarella and Fresco) with a long record of scientific papers on this topic and also with ample experience in the policy arena. So we feel that there is enough experience to justify the statements made.

Line 389-390. Do you have a reference for the widespread indiscriminate us of ptf? You should admit too that often there is no other choice than using them. Answer: we have added two references for the ptfs.

Lines 413-414. This is mainly true for cultivated soils and grasslands, on the other hand, in many forested parts of the world, organic carbon accumulates because there is no biological activity and increases acidity. The carbon pool we need for biodiversity is a "living" carbon pool, not accumulation of acid O layers. Answer: We have added a sentence to explain that we only mean soils with active living carbon pools. Good point. . Lines 419-421. I think you are very optimistic about monitoring changes in SOC with sensors. Given the error of measurement they have by now, and given the rate of change in SOC, it will take nearly 50 or 100 y to prove a change. Moreover SOC are not so relatively easy to measure and to monitor given their high short-scale spatial variability. Answer: We agree with the reviewer that measuring SOC is a complex issue in terms of measurement and of spatial heterogeneity. Although we are fully aware of this, we would like to keep the statements as made as it provides a guiding point for the future making organic matter a rough proxy for soil quality. Recent papers on sensing, as cited, indicate good possibilities for measuring SOC so we feel that this, in time, may help to obtain a sufficient number of observations.

Same for lines 368-370, for remote sensors it is even worse because of atmospheric effects, vegetation cover, etc. Answer: Yes we fully agree, but see answer above. Use of proximal sensors may have more potential.

Same again for lines 430-431. "Preferably by remote sensing" this is a very dangerous assumption. First it is not yet operational at all, second, if I am a funding agency, I will never give you again one cent for real measurements but tell you, okay, just look at the satellites images, some of them are free ! Answer: Yes, we agree with this pragmatic point of the reviewer and we have removed this statement from the text.

Recommendation. I'm surprised not to find anything about data collection and data sharing. Answer: We added a sentence about this in the recommendations. Thank you. Good point.

Throughout the text, I'm also surprised that there is not ref to the recent reports delivered by the ITPS, and even not to paper by Montanarella et al on this and in this journal. Answer: we have added some sentences about the recent paper in SOIL ( which covers ITPS) and the new Nature communication paper, both by by Luca Montanarella et al. .

Refs The paper by Montanarella in Nature is published, please add issue and lines numbers. Answer: we have updated the reference Overall a nice paper for the FORUM. Answer: Thanks!